# Geometric deep learning improves generalizability of MHC-bound peptide predictions
Dario F. Marzella [1,5], Giulia Crocioni [2,5], Tadija Radusinović [3], Daniil Lepikhov [1], Heleen Severin[1], Dani L. Bodor [2], Daniel T. Rademaker [1], ChiaYu Lin[2], Sonja Georgievska [2], Nicolas Renaud [2], Amy L. Kessler [4], Pablo Lopez-Tarifa[2], Sonja I. Buschow [4], Erik Bekkers [3] & Li C. Xue [1] ✉

The interaction between peptides and major histocompatibility complex (MHC) molecules is pivotal in autoimmunity, pathogen recognition and tumor immunity. Recent advances in cancer immunotherapies demand for more accurate computational prediction of MHC-bound peptides. We address the generalizability challenge of MHC-bound peptide predictions, revealing limitations in current sequence-based approaches. Our structure-based methods leveraging geometric deep learning (GDL) demonstrate promising improvement in generalizability across unseen MHC alleles. Further, we tackle data efficiency by introducing a self-supervised learning approach on structures (3D-SSL). Without being exposed to any binding affinity data, our 3D-SSL outperforms sequence-based methods trained on ~90 times more data points. Finally, we demonstrate the resilience of structure-based GDL methods to biases in binding data on an Hepatitis B virus vaccine immunopeptidomics case study. This proof-of-concept study highlights structure-based methods' potential to enhance generalizability and data efficiency, with possible implications for data-intensive fields like T-cell receptor specificity predictions.

Peptide-major histocompatibility complex (pMHC) interactions play a key role in the immune surveillance system, as T-cell discrimination between self and non-self relies on T-cell receptor (TCR) recognition of peptides presented by Major Histocompatibility Complex (MHC) molecules. The accurate identification of peptides presented by MHC on the cell surface is crucial for understanding autoimmune diseases[1], recognizing pathogens[2], and addressing transplant rejection[3]. The recent notable clinical advancements in cancer immunotherapies[4,5], specifically targeting tumor-associated or tumor-specific peptide-MHC complexes, underscore the urgency to advance computational methods for identifying MHC-bound peptides[6,7]. With over 14,000 (according to the IPD-IMGT/HLA database[8]) human MHC-I proteins encoded by the canonical human leukocyte antigen (*HLA*) genes (*HLA-A*, *HLA-B*, and *HLA-C*), the theoretical $20^9$ 9-residue peptides (called 9-mers) create impractical in vitro testing scenarios. This calls for the development of in silico peptide-MHC (pMHC) binding prediction methods.

Much effort has been devoted to design predictors for pMHC binding[9]. Most state-of-the-art (SOTA) predictors are sequence-based machine learning methods[10–12] trained on a large amount of experimental binding data for pMHC (~600 K for MHC-I[11] and >480 K for MHC-II[10]). Sequence-based (SeqB) methods take amino acid sequences as input to predict binding affinity (BA) or eluted ligand rank. In addition to this, NetMHCpan uses a structure-derived concept like the MHC pseudosequence[13], utilizing only the MHC residues observed to interact with the peptide in X-ray structures instead of the whole MHC sequence. MHCflurry 2.0 adds to the MHC-I binding prediction an antigen processing model, integrating both scores in its final MHC presentation predictor. These methods are reported to perform well on most of 214 well-studied alleles with hundreds of thousands of mass spectrometry data.

While these approaches have shown considerable advancement and contributions to clinical trials[14,15], they are not without limitations and have been shown to provide highly discordant predictions[16,17]. A notable constraint stems from their reliance on extensive data, posing challenges for the thousands of less-explored HLA alleles with limited available binding information. Moreover, redundancy reduction strategies, such as the

[1]Medical BioSciences department, Radboudumc, Radboud University Medical Center, 6525 GA Nijmegen, The Netherlands. [2]Netherlands eScience Center, Amsterdam, The Netherlands. [3]University of Amsterdam, Amsterdam, The Netherlands. [4]Department of Gastroenterology and Hepatology, Erasmus MC, University Medical Center Rotterdam, 3015 GD Rotterdam, The Netherlands. [5]These authors contributed equally: Dario F. Marzella, Giulia Crocioni.
✉e-mail: li.xue@radboudumc.nl

removal of identical peptides or using independent test sets[11,12,18], are not consistently implemented. This lack of emphasis on ensuring dissimilarity between test and training data can contribute to an overly optimistic performance. Finally, these methods typically train and test on a small subset of MHC alleles, which is a limiting factor considering the broad spectrum of human alleles. Consequently, predictions for underrepresented alleles in these sets may exhibit lower accuracy, akin to an out-of-distribution (OOD) issue in machine learning, where real-case scenarios differ significantly from training data, introducing challenges in generalization[19–21].

A structure-based (StrB) method can have several compelling advantages over SeqB methods[22] as it can 1) integrate the huge amount of experimental pMHC BA data with physico-chemical properties of 3D interfaces, 2) reflect minor changes of mutations in 3D space and energy landscape, and 3) naturally handle variable peptide lengths in 3D space. The booming advances of geometric deep learning (GDL)[23–27], a specialized branch tailored for 3D objects, underscore the timely opportunity for enhancing pMHC binding predictions through the development of StrB methods. Importantly, the highly conserved MHC structures observed across various species, together with the constrained peptide binding within the MHC binding groove, establish a foundation for constructing accurate pMHC 3D models[28,29]. These models, in turn, serve as reliable input data for GDL algorithms.

Much efforts have been devoted to design StrB methods for predicting MHC-binding peptides with different degrees of success in the past decades[22]. Typically, energy terms, statistical potentials and structural descriptors (e.g., the number of polar-polar interactions) are used as input of machine learning algorithms[22]. Based on 77,000 3D models generated with APE-Gen[29], the authors of 3pHLA[30] used Rosetta[31] to extrapolate energy scores at each peptide position and provided these scores as vectors to train a random forest predictor. Recently, Wilson et al.[32] have developed a StrB binding motif prediction method, for which they trained inception convolutional neural networks (CNN) on electrostatic potentials of MHC structures, demonstrating high prediction speed and precision. Finally, MHCfold predicts pMHC 3D structures and the interactions at the same time and achieved promising results in both tasks[33].

In this study, we develop end-to-end GDL methods that directly analyze structural features at atom or residue level to predict the binding between peptide and MHC. We set out to demonstrate the generalizability issue for pMHC binding predictions and evaluate whether our GDL-based StrB methods have a more robust generalization than SeqB methods for pMHC binding predictions. We compared three different GDL StrB networks with two SeqB networks on an allele-clustered scenario, deliberately minimizing the similarity between the test and training data. To assess the models' performance, we compared these results with those obtained from a shuffled dataset, enabling an evaluation of the accuracy drop when extrapolating to unseen cases. In addition, being inspired by the success of the self-supervised learning (SSL) in natural language processing[34,35], we introduced a self-supervised GDL method, 3D-SSL, novel to the best of our knowledge. Remarkably, this method showed superior potential in data efficiency even without exposure to any BA data. Finally, we demonstrate the robustness of StrB methods against biases in the binding data through a study-case on *HBV* vaccine design. This study underscores the potential strength derived from the integration of 3D physics-based modeling and data-driven geometric deep learning to enhance both generalizability and data efficiency. These findings hold broad implications for the fields of immunology and therapy design.

## Results
### Data separations using allele clustering
We hypothesize that structure-based predictors may be more generalizable to unseen alleles than sequence-based approaches due to ultra-conserved MHC structures. To evaluate this hypothesis, we use ~100 K pMHC BA data covering 110 HLA alleles and peptides lengths spanning between 8 and 21 residues (Fig. 1A, see details in Methods). We tested the StrB over SeqB approaches on unseen alleles using two data configurations: a randomly

shuffled data configuration as the control baseline, resembling the SOTA-used data separation schemes, and an allele-clustered configuration to evaluate their generalization power. The allele-clustered data configuration was designed to evaluate the applicability of the neural networks to a test set composed of unseen data, the distribution of which might deviate significantly from the training data (e.g. patients with less-studied alleles). For all configurations, pMHC 3D models were processed and featurized using various software tools to generate encoded sequences, 3D grids, or graphs, depending on the approach. For a comprehensive overview of the featurized data characteristics, including feature size, encoding, and representation, refer to the Methods section (Tables 1–3).

In the shuffled configuration, we randomly split the data in training, validation and test sets, using target stratification to ensure similar target distribution in all data sets (Fig. 1A). Binders and non-binders were represented at 44% and 56% in each set, respectively. This configuration aligns with common practices found in existing literature, where the test set is typically either randomly extracted from the initial dataset or is an independent dataset sharing similar alleles and peptides distributions[11,18]. In the second experiment (allele-clustered), we clustered the data based on the MHC pseudosequence[13] using a PAM30-based[36] hierarchical clustering for each gene, i.e., *HLA-A*, *-B* and *-C*, respectively (Fig. 1B-D, see also Methods). Then, from a total of ~100 K BA data (see details in Methods), for each gene we selected the most distant clusters based on the MHC pseudosequence's dendrogram so that about 10% of each gene's data (10% for *HLA-A*, 11% for *-B* and 12% for *-C*, respectively) was selected as test set. We split the remaining data into training and validation sets in a target-stratified manner. This design allows the networks to sample a large number of cases from each HLA gene, while strategically incorporating only distant clusters of alleles into the test data. This configuration aims at simulating real-case scenarios in which the network has to provide a prediction on alleles it has never seen or seen infrequently. We build 3D models for all these BA data using PANDORA[37], which served as input of our GDL approaches (Fig. 1E).

### StrB predictors demonstrate greater generalizability than SeqB on distant alleles
For StrB predictors we used different network architectures with different types of physico-chemical features (Fig. 2A). We developed an atom-level 3D-CNN and a residue-level graph neural network (GNN) using different structural and energy-related features (Tables 1 and 2), and a residue-level E(n) equivariant GNN (EGNN) using as input features only the residues identities and their xyz coordinates (Table 3). 3D-CNNs analyze structural features that are mapped on a 3D grid, capturing local geometric relationships effectively. GNNs and EGNNs treat structures as connected nodes (representing atoms and/or residues) on a graph, updating the features of a node iteratively through passing information of its neighboring nodes[38]. In handling 3D structures, a key requirement for GDL is rotation-translation (RT) invariance, meaning consistent predictions regardless of input orientation. 3D-CNNs capture local geometric relationships but may be sensitive to rotations. GNNs ensure constant outputs even with input rotations. EGNNs further enhance its networks by enabling both the output and all of its features of hidden layers to consistently rotate as well. EGNNs, known for simplicity, are widely used in predicting molecule properties and generating realistic small molecules[24,26,39]. Since many SOTA SeqB predictors are not available to be re-trained for a fair comparison, we developed a multilayer perceptron (MLP) to represent a baseline sequence-based approach. We also re-trained MHCflurry2.0, being a SOTA architecture available for re-training. Considering that MHCflurry2.0 is an ensemble predictor made of multiple MLPs, we designed our MLP to use one representative architecture of individual MLPs in MHCflurry2.0, in order to assess the single MLP contribution compared to the ensemble effect on the predictions performances. Moreover, our MLP serves the purpose of showing how simple it can be for a SeqB method to achieve high performances in a shuffled set configuration, and that this does not guarantee similar performances on unseen alleles.

As expected, in the shuffled data configuration, all methods perform well (Fig. 2B), with the MLP and our EGNN showing a slight edge over the others, with area under the ROC curve (AUC) of 0.91 and 0.87, respectively.

See area under the precision-recall curve (AUPRC) details in Supplementary Note 1. SeqB methods learn the features distribution and can provide good predictions, as suggested from their good performance shown in

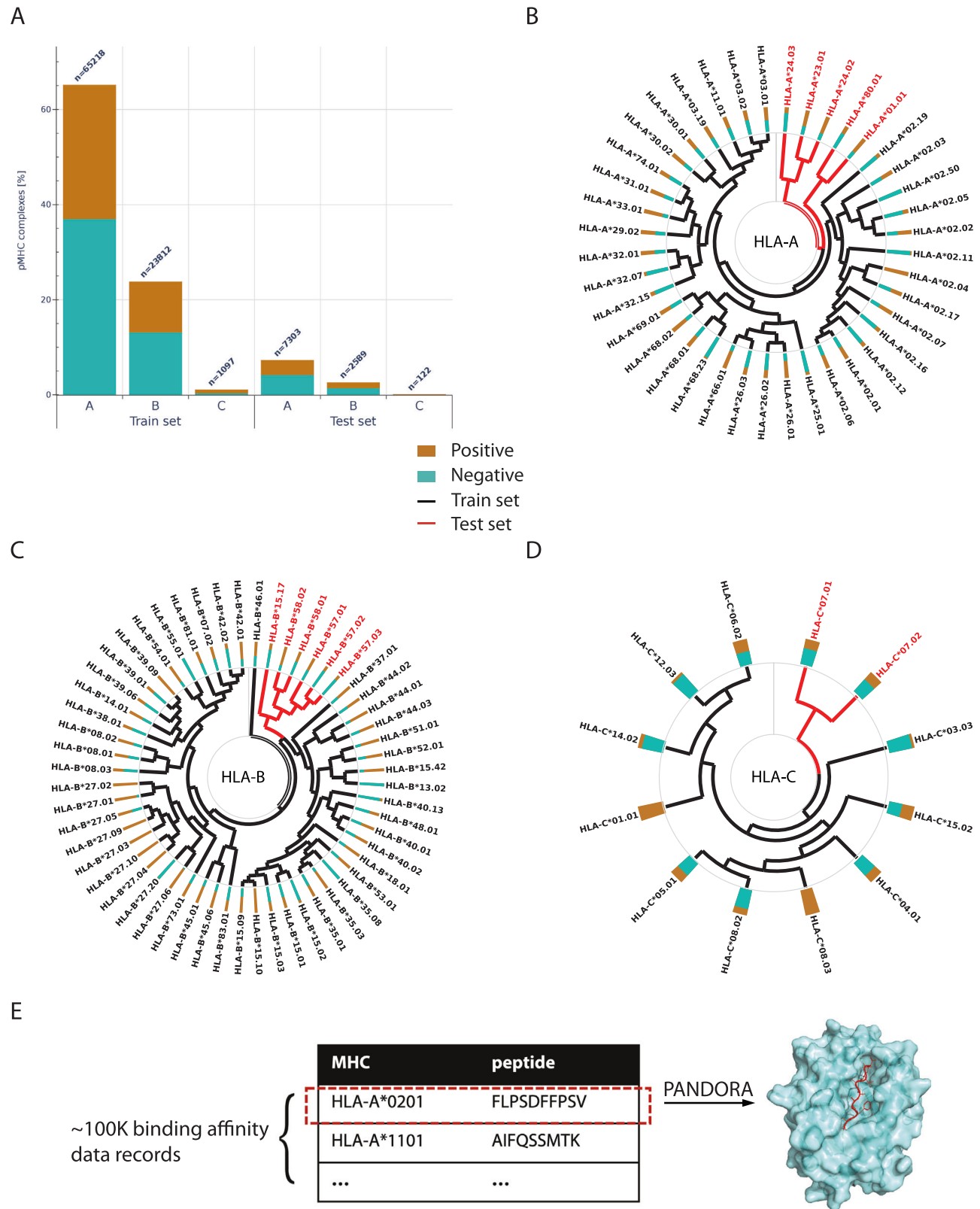

**Fig. 1 | Data overview. A** Shuffled data distribution representation between HLA-A, -B and -C for the training and test sets. **B-D** show the hierarchical clustering of HLA-A, -B, -C pseudosequences, respectively, with training data in black and test data in red. In all panels, the mustard/teal bar by each gene label represents the binders/non-binders (positive/negative) ratio for the gene. **E** Data enrichment through 3D modeling. We used PANDORA[37] to generate 20 3D models of each of the 100178 pMHC data points, thus enriching the sequence information with physics-derived information, such as geometry and physico-chemical features.

Fig. 2B. StrB methods' high performance suggests their potential in reaching performance comparable to the current SeqB literature.

In the allele-clustered configuration (Fig. 2C), all predictors show a decline in performance, highlighting the generalizability issue. StrB approaches demonstrated more robust performance than SeqB methods on distant alleles. StrB consistently outperforms SeqB by 5–11% in terms of AUC. The substantial performance drop of SeqB methods underscores their susceptibility to the out-of-distribution issue, highlighting the importance of rigorous evaluation for obtaining real-life estimations of predictive methods. MHCFlurry is an ensemble predictor using multiple neural networks, demonstrating better generalization than a basic MLP. Figure 2D shows the performances on each test allele ordered with increasing sequence evolutionary distance to the training alleles. Computational efficiency for the StrB methods is reported in Supplementary Note 2.

Notably, EGNN, our simplest GDL method and the top-performing StrB, achieves a significantly better AUC and AUCPR compared to SeqB methods by 8–11% (Fig. 2C, Supplementary Fig. 1). EGNN, using only amino acid types and distances, virtually distinguishes itself from the SeqB methods solely by integrating the spatial relative position of residues and the network's treatment of this information. This suggests that the enhanced generalization ability of our StrB methods relies on the inclusion of geometrical information in the 3D models and the GDL's capacity to incorporate such details in the learning process, rather than being dependent on a specific set of features or network architecture. Note that the AUPRC values reported here may differ in real-world scenarios with highly imbalanced data, where negative instances significantly outnumber positive ones. Our goal is to demonstrate the relative generalization of geometric deep learning over sequence-based methods, though performance may vary with dataset imbalance.

### Self-supervised learning demonstrates superior data efficiency

Self-supervised learning is a game-changing technique for natural language processing (NLP)[34,35]. Many well-known architectures, including BERT[34], GPT-X[35], MAEs (Masked Autoencoders)[40] are SSL at their core. SSL also plays an important role in AlphaFold, a revolutionary AI-based protein structure predictor[25]. Unlike supervised prediction, where a network is trained to directly predict a target value (a binding label, in our case), in SSL we mask or noise part of the data, and task the network with unmasking/denoising the input, for example, training the network to predict the masked word identity from a sentence or masked amino acid identity from a structure. Because of this self-prediction nature, SSL does not require any BA data, thus is potentially data efficient. Here we extend SSL to 3D and evaluate its performance on pMHC binding predictions.

*Training.* Our 3D-SSL method is trained on a masked residue prediction (MRP) task (Supplementary Fig. 2). At training time, a random 20% of the residues in the input structure is masked each epoch, i.e. set to a dedicated 'unknown' token, and every masked residue constitutes a separate input datapoint. In this way, every residue in each pMHC structure (around 192 residues) is essentially treated as a separate datapoint, effectively increasing the training set. The EGNN network is then trained to predict the probability of the 20 types of amino acids for each masked residue based on its spatial neighbors.

*Inference.* Once the network has been trained on X-ray structures, we use it to predict the peptide binding of our test allele-clustered dataset. The trained network takes a pMHC 3D model as input, masks each residue in the peptide and predicts the probability of the 20 amino acid types at the masked position. The key innovation consists then in converting the predicted probabilities into the binding energy a statistical potential using a Boltzmann distribution. In computational structural biology, Boltzmann distribution is widely used to approximate energies (known as "statistical potentials") for predicting protein folding and computational docking[41–43]. In Boltzmann distribution, the frequency of an observed state $i$ is proportional to its energy $e_i$:

$$p_i \propto e^{-\frac{\varepsilon_i}{kT}}$$

where $k$ is the Boltzmann constant and $T$ is the temperature.

## Table 1 | Input features to the CNN architecture

| Feature and shape and type | Description |
|---|---|
| SkipGram [6, 1], float | Skip-Gram embedding for residue type[72] |
| Edesolv [1], float | Desolvation energy, calculated as in HADDOCK[71]. |
| Anchor [1], bool | Label to indicate if the residue is an anchor residue (peptide side) or a pocket residue (MHC side) |
| RCD_apolar-apolar [1], float | Residue Contact Density for apolar-apolar interactions calculated as in Vangone and Bonvin[78] |
| RCD_apolar-charged [1], float | Residue Contact Density for apolar-charged interactions calculated as in Vangone and Bonvin[78] |
| RCD_charged-charged [1], float | Residue Contact Density for charged-charged interactions calculated as in Vangone and Bonvin[78] |
| RCD_polar-apolar [1], float | Residue Contact Density forpolar-apolar interactions calculated as in Vangone and Bonvin[78] |
| RCD_polar-charged [1], float | Residue Contact Density for polar-charged interactions calculated as in Vangone and Bonvin[78] |
| RCD_polar-polar [1], float | Residue Contact Density for polar-polar interactions calculated as in Vangone and Bonvin[78] |
| RCD_total [1], float | Residue Contact Density for total interactions calculated as in Vangone and Bonvin[78] |
| bsa [1], float | Buried Surface Area |
| charge [1], float | Charge based on the OPLS force field implemented in HADDOCK[71] |
| coulomb [1], float | Electrostatics based on the OPLS force field implemented in HADDOCK[71] |
| vdwaals [1], float | Van Der Waals energy based on the OPLS force field implemented in HADDOCK[71] |

Full features details on built-in deeprank features can be found at https://deeprank.readthedocs.io/en/latest/deeprank.features.html.

In traditional statistical potentials, the state $i$ often is a residue-residue contact pair (two-body). Intuitively, if a residue-residue pair is often observed in the Protein Data Bank (PDB) structures (i.e., has a high frequency), such interactions are believed to have low energies (the hallmark of native conformations).

In this work, we convert the predicted probabilities from 3D-SSL into the energy contribution using Boltzmann distribution: $e_i = -\log(p_i)$ (for simplicity, we ignored the reference state and set kT=1 here). In essence, the 3D-SSL is predicting multi-body statistical potentials. The energy contributions of each peptide residue are then summed up as the binding energy of the whole pMHC structure (Fig. 3A, Methods). We used this final score as a prediction for the BA. Each residue's contribution is treated as an independent event, meaning the conditional probabilities of each residue are considered in a non-autoregressive manner.

Our 3D-SSL is a surprisingly efficient learner. For a proof-of-concept, our 3D-SSL is currently trained only on 1,245 X-ray structures from TCRpMHC complexes (Fig. 3D, Methods). On the allele-clustered dataset, 3D-SSL, trained only on ~1 K experimental complexes and not being exposed to the BA values, already outperformed the SeqB supervised methods trained on >90 K BA data (Fig. 3B). This result shows the superior data efficiency potential of SSL networks.

To further estimate the data efficiency of 3D-SSL against StrB methods, we trained the supervised EGNN on small subsets of the BA training set to estimate the amount of 3D models needed for the supervised network to outperform the self-supervised learning on X-ray structures. When comparing the supervised EGNN against 3D-SSL on the allele-clustered set (Fig. 3C) we can see the EGNN requires approximately 10,000 BA data points to consistently outperform 3D-SSL, which achieves comparable performance with training on only 1245 complexes, roughly eight times less data. This indicates that if many high-quality pMHC structures with BA

**Table 2 | Input features to the GNN architecture**

| Feature and shape and type | Description |
|---|---|
| NODE FEATURES | |
| res_type [21, 1], bool | One-hot representation of the node's residue (20 amino acids + unknown). |
| res_size [1], int | The number of non-hydrogen atoms in the side chain. |
| res_mass [1], float | The average residue mass in Da. |
| res_charge [1], int | The charge of the residue in fully protonated state in Coulomb. |
| res_pl [1], float | The isoelectric point, i.e., the pH at which the molecule has no net electric charge. |
| polarity [4, 1], bool | One-hot representation of the polarity of the amino acid: NONPOLAR, POLAR, NEGATIVE, POSITIVE. |
| hb_donors [1], int | The number of hydrogen bond donor atoms in the residue. |
| hb_acceptors [1], int | The number of hydrogen bond acceptor atoms in the residue. |
| res_depth [1], float | The average distance in Å of the residue to the closest molecule of bulk water, computed via BioPython. |
| hse [3, 1], float | Half sphere exposure, which indicates how buried an amino acid residue is in the biomolecule, computed via BioPython. |
| sasa [1], float | Solvent-accessible surface area, in Å², computed via FreeSASA. |
| bsa [1], float | Buried surface area, which represents the area of the complex interface, in Å², computed via FreeSASA. |
| irc_total [1], int | The number of residues on the other chain that are within a cutoff distance of 5.5 Å. |
| irc_negative_negative irc_negative_positive, irc_nonpolar_negative, irc_nonpolar_nonpolar, irc_nonpolar_polar, irc_nonpolar_positive, irc_polar_negative, irc_polar_polar, irc_polar_positive, irc_positive_positive [1], int | As above, but for specific residue polarity pairings. |
| EDGE FEATURES | |
| same_chain [1], bool | Boolean indicating whether the edge connects nodes belonging to the same chain (1) or separate chains (0). |
| covalent [1], bool | Boolean indicating whether nodes are covalently bound (1) or not (0). Covalency is not directly assessed, but any edge with a maximum distance of 2.1 Å is considered covalent. |
| electrostatic [1], float | Electrostatic potential (also known as Coulomb potential) between two nodes, calculated using interatomic distances and charges of each atom. |
| vanderwaals [1], float | Van der Waals potential between two nodes, calculated using interatomic distances and a list of atoms with Van der Waals parameters. |
| distance [1], float | Interatomic distance between atoms in Å, computed from the xyz atomic coordinates taken from the PDB file. |

Full features details can be found at https://deeprank2.readthedocs.io/en/latest/features.html.

measurements are available for a given problem, supervised StrB methods will likely result in better predictive performance. However, in a low-data regime, e.g., working with rare alleles or novel neoantigens, training self-supervised models could likely perform better than supervised ones. It is important to note that when training 3D-SSL purely on 3D models instead of X-ray structures, it shows a considerably lower performance (AUC of 0.56 ± 0.01, Supplementary Note 3). This indicates that our self-supervised approach needs high-quality structures.

Interestingly, we noticed that including peptide-TCR structures in the training set of 3D-SSL improved the prediction for pMHC BA. We have tested two different training sets for 3D-SSL: one including only pMHC structures and one including also peptide-TCR (T-Cell Receptor) structures, extracted from full TCR-peptide-MHC X-ray complexes. We found out that the larger dataset including peptide-TCR structures provided higher BA prediction performance, from AUC of 0.640 ± 0.011 to 0.668 ± 0.005

(Fig. 3D). The improved results with peptide-TCR data might indicate that adding competing binding environments contributes to the prediction of the BA between peptide and MHC.

## A case study: vaccine for chronic *Hepatitis B Virus* (*HBV*) infection

We compared our best StrB method, the EGNN (trained on ~100 K BA data), to two SOTA softwares (close to a million combined BA and mass spectrometry data) on an *HBV* vaccine design study[44]. In this study, *HLA-A\*02:01* matched dendritic cells (DCs) from healthy donors were exposed to synthetic long peptides containing *HBV* sequences in order to test which peptides would be presented, and therefore would serve as possible effective vaccine candidates (Fig. 4A). Notably, two closely related peptides (HLYSHPIIL, denoted as HL, and HLYSHPIIL**G**, denoted as HG, differing only in an additional C-terminal glycine in HG) were experimentally

## Table 3 | Input features to the EGNN architecture

| Feature and shape and type | Description |
|---|---|
| **NODE FEATURES** | |
| res_type [1], int | Index of node residue type. |
| pos [3], float | The xyz coordinates of the residue C-alpha. |
| entity [1], bool | Boolean indicating whether the node is the peptide (1) or the protein (0). Used during pooling to identify which residues outputs to sum together (i.e. the ones with entity=1, corresponding to the peptide). |
| **EDGE FEATURES** | |
| same_chain [1], bool | Boolean indicating whether the edge connects nodes belonging to the same chain (1) or separate chains (0). |

confirmed as binders through mass spectrometry and an in vitro HLA binding assay[44]. However, only HL but not HG is predicted as a binder by the NetMHCpan4.1b web server and the BA predictor of MHCflurry 2.0's official release (Fig. 4B). The HL peptide has been reported numerous times in literature as an *HLA-A*02* binder[45–47], thus it is present in the training set of NetMHCpan4.1, MHCflurry 2.0 and our EGNN. The HG peptide instead is not present in any of the training sets to the best of our knowledge, as it was reported for the first time in Kessler et al.[44]. In a blind test setting (i.e., we did not know the labels of binding or not beforehand when running our StrB methods), we went on to check whether our StrB methods can accurately identify the HL/HG peptides as binders. We first trained our EGNN on all the data available in our dataset, splitting it in train and validation only without any test set, in order to use as much data as possible for training. Then, we generated 3D models of the pMHC complex between the HL/HG peptides and *HLA-A*02:01*, and our trained EGNN networks to make separate predictions. Both of our networks were able to predict both peptides as binders with high confidence scores (Fig. 4B).

When inspecting their 3D models it becomes apparent that the HG/HL case is in fact a relatively easy case for structure-based approaches. The C-terminal glycine is overhanging without causing major unfavorable interactions or clashes (Fig. 4C, see also Supplementary Video 1). The HL peptide can form a weak salt bridge with MHC LYS146 using its C-terminal carboxyl group. The addition of the glycine in HG causes a bigger entropic effect (i.e. the glycine has to reduce its entropy to fit in the pocket), which can be balanced by the interactions of the glycine's carboxyl group, which takes a weak salt bridge with LYS146 and an extra hydrogen bond with THR80. Overall, this result indicates that structural modeling adds an important biochemical environment to assist structure-based AI to discover the unconventional HG peptide.

It is critical to emphasize that the biases in SeqB networks might result from training data with potential bias. In fact, the proteasome, which is responsible for peptide generation in non-antigen presenting cells (APC) cells, has a low probability of cleaving the C-terminal bond of a glycine[48]. As a result, C-terminal glycines, presumably generated by endosomal proteases in APCs, are poorly represented in training sets which contain mostly peptidomes of parenchymal cells/ non-APCs, likely causing the poor performance of SeqB methods on peptides with C-terminal glycines. MHCflurry 2.0 addresses similar challenges with its antigen preprocessing (AP) module[11], which could filter out peptides not supposed to be processed by non-APC cells[49,50], such as peptides containing C-terminal glycines. However, this does not justify the inaccuracies in their BA predictor, which should be able to predict these peptides as binders when presented with them. Overall, the robustness of the StrB EGNN might prove effective against other issues caused by the biases that SeqB networks inherit from the data, such as the low accuracy in predicting MHC binding of cysteine-containing peptides, caused by underrepresentation of cysteines in the MS-derived immunopeptidomes used for training.

## Discussion

Generalizability is an important topic in a broad range of studies[51–53]. In this proof-of-concept study, we investigated the generalizability of various StrB approaches in predicting peptide-MHC interactions, a pivotal aspect of immune surveillance and a major bottleneck in the design of cancer vaccines[54–56] and TCR therapies[57,58]. We showed that our StrB networks have greater generalization power compared to conventionally used SeqB networks when evaluated on unseen MHC alleles. Interestingly, our top StrB method EGNN, utilizing only amino acid types and distances, distinguishes itself from SeqB mainly for the incorporation of spatial positions. This indicates that the enhanced generalization in our StrB methods relies on GDL's capacity to integrate 3D geometric information during learning, rather than specific features or network architecture. A similar trend can be seen in our self-supervised approach, 3D-SSL, which uses the same features as our supervised EGNN. Notably, 3D-SSL outperforms SeqB approaches trained on ~90 times more data, demonstrating its potential data efficiency, being trained on minimal data and features. The presented HBV test case, although not representing an extensive comparison of SeqB and StrB neural networks biases, exemplifies how StrB methods can better capture the biochemical environment of the pMHC complex, potentially minimizing non-biochemical biases in the data.

We demonstrate the efficiency and feasibility of boosting experimental BA data with physics-derived pMHC 3D models to achieve better predictions of MHC-bound peptides. The inherent conservation of MHC structures makes large-scale physics modeling feasible. These models offer essential complementary information to biochemical binding assays, aiding in training highly generalizable algorithms for the identification of MHC-bound peptides. We anticipate that the fusion of 3D modeling with the expanding capabilities of GDL will soon provide robust and valuable tools for prioritizing targets in cancer immunotherapies, thereby accelerating development timelines and significantly reducing associated costs. Furthermore, MHC structures exhibit strong conservation, with peptides confined to the binding groove, making them an ideal toy model for AI research, particularly in geometric deep learning. The rich experimental binding data, combined with our 3D models, holds potential to advance AI algorithms, providing novel solutions and insights for both the AI and immunology communities.

While our study is on pMHC-I, our StrB methods are directly applicable to pMHC-II complexes, which play a pivotal role in cancer immunotherapy responses[59,60] but their BA is challenging to predict. In fact, MHC-II have an open-ended binding cleft that allows for longer peptides to bind, thus including peptide flanking regions (PFRs) which are highly variable in length and can affect the pMHC BA[61]. StrB methods can naturally handle peptide length variability in 3D space while SeqB methods are inherently limited as they take fixed length input. All together, our StrB approaches, especially, our 3D-SSL approach, may provide a data efficiency solution for data hungry fields, for example, the long-standing challenge of TCR specificity predictions[62]. However, 3D-SSL is trained and tested on pMHC complexes alone and its impact on other protein-protein interfaces (PPIs) needs further extensive investigations. Expanding the 3D-SSL information source to the Protein Data Bank[63] structures and AlphaFold2[25,64] models and integrating supervised methods with our 3D-SSL are potential next steps.

## Methods

### Binding affinity data collection

BA data was collected from O'Donnel et al.[11] through MHCflurry 2.0 "'download-fetch'" command, which yielded a total of 598,450 pMHC BA measurements. We selected only qualitative essays on human alleles and filtered out data points considered non-exact values (i.e. samples assigned to have a value lower or higher than a certain threshold, with no exact value associated). The final dataset consists of 100,178 data points each containing a peptide sequence, an MHC allele name and their measured BA, using a threshold of 500 nM to separate binders from non-binders, resulting in 44,102 binders and 56,076 non-binders from 110 HLA alleles with lengths varying from 8 to 21 residues (see Fig. 1 and Data and Code Availability).

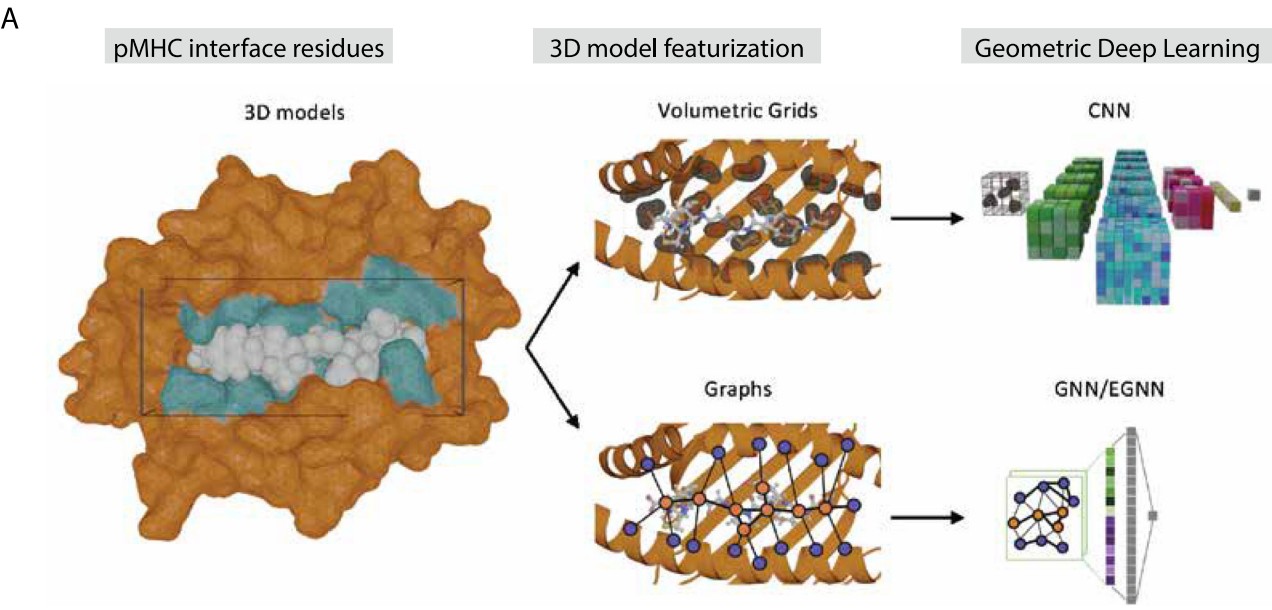

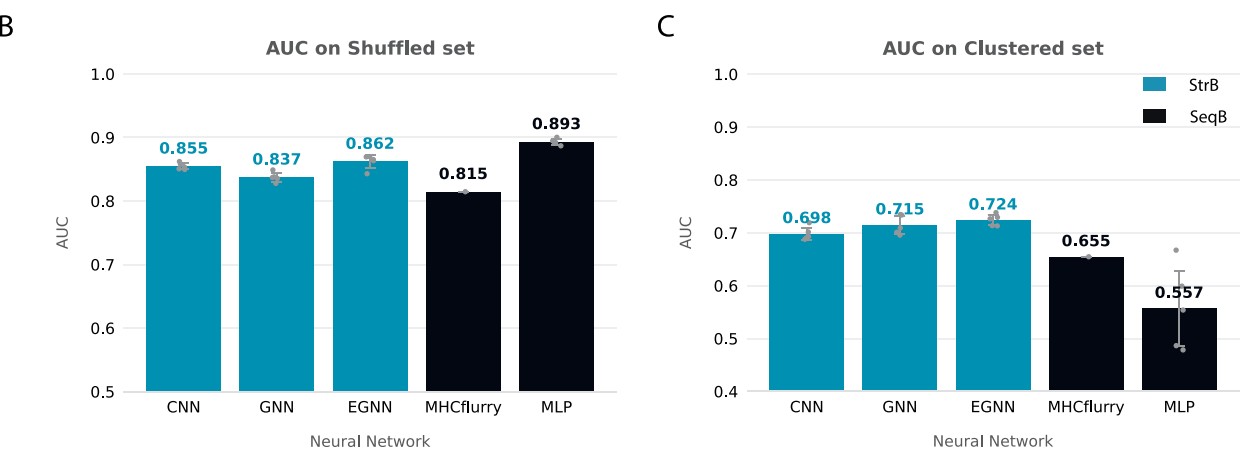

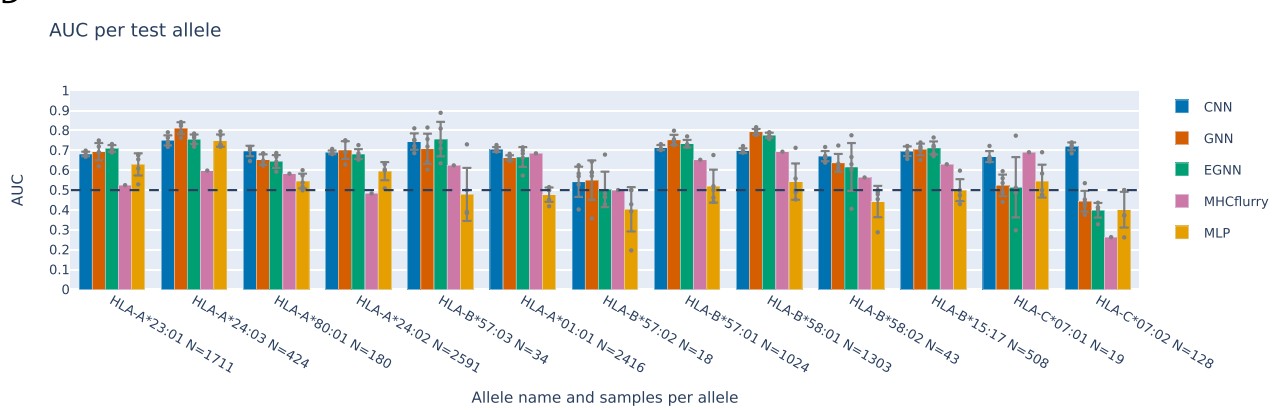

Due to issues in the data processing and data featurization, some networks could use a few datapoints less than the total dataset. The final dataset size per model was: 100,178 for CNN and EGNN; 100,069 for GNN; 100,090 for MHCflurry and MLP. As the difference in datasets amounts for about 0.1% of the data and it is proportionally distributed in train and test sets, we expect it to not influence the results.

### 3D modeling with PANDORA

The BA dataset was processed with PANDORA (v2.0beta2) software[37] to generate 3D models of the pMHC complexes for these BA data. For every case, PANDORA uses NetMHCpan4.1[10] to predict the peptide's binding core, using it to select the anchor residues to add restraints. The 3D models were generated with PANDORA's fully flexible mode (i.e., 20 3D models per

**Fig. 2 | Our supervised StrB methods overview and performances. A** Overview of the pipeline employed for processing pMHC complexes and running the supervised structure-based networks. The process involves: 1) Identifying interface atoms and residues from the pMHC 3D models, 2) converting each pMHC interface into volumetric grids and graphs enriched with geometric and physico-chemical information (Tables 1–3), 3) Run the networks on these representations: volumetric grids for the CNN and graphs for GNN and EGNN. GNN and EGNN use similar graph topology, differing on features and message passing framework (see details in Methods). **B, C** The performance of StrB and SeqB methods on the shuffled test dataset and on the allele-clustered test dataset, respectively. For all the networks

except MHCflurry, 5 replicas were performed by randomly re-sampling each validation set, and the error bars show the standard deviation between the five replicas. Each replica is marked as a gray dot. MHCflurry handles the separate validation sets internally and collects the networks' outputs, as such no standard deviation could be retrieved. **D** AUC per allele on the allele-clustered test set. Allele name and number of test cases are reported on the x-axis, and the alleles are sorted by sequence distance with the training set. The black dashed line marks the random predictor AUC value of 0.5. Error bars show the standard deviation between the five replicas. Each replica is marked as a black dot.

case, with a restraints flexibility allowed of 0.3 Å), with the exception that only the G-domain of the MHC is modeled, while the rest of the MHC α-chain and the β-2 microglobulin are ignored. The top molpdf[65] scored model was selected as the best model and used as the training, validation and test data.

### Data clusterings

In the shuffled data configuration, we split the data into training (90%) and test (10%) sets after random shuffling, stratifying them on the binary target to make sure the positive and negative cases are equally represented in the different sets. During the training phase, 15% of the training set was used as a validation set.

For the allele clustering, allele pseudosequences were collected from O'Donnel et al.[11]. As explained in Hoof et al.,[13] these pseudosequences only contain the residues at the MHC positions that have been shown in X-ray structures to interact with the peptide, which makes it so every pseudosequence has the same length and no alignment is needed, as every position correspond to the same physical position on the MHC structure. The pseudosequence were separated per gene and scored against each other using a PAM30 scoring matrix, and the so-obtained evolutionary distances were used to build a hierarchical clustering using the cluster.hierarchy module from scipy[66] for each HLA gene. From these three clusterings (one per gene), we selected the most distant clusters until reaching the closest possible value to 10% of the data (10% for *HLA-A*, 11% for *HLA-B* and 12% for *HLA-C*) to form the test set. All the other data points were left as a train/validation set. Note that the dataset includes 37 *HLA-E* cases from two alleles, making clustering unfeasible, thus they were included in the training/validation set.

### Training data for 3D-SSL

We extracted 1245 entries by overlapping PDB IDs containing MHCI from ProtCID[67] with the protein-peptide database Propedia[68]. These Propedia entries are composed of 517 pMHC and 731 TCRα-peptide or TCRβ-peptide structures, separated out from the original PDB structures. We consider two SSL datasets from these entries. First, a reduced dataset of 517 only pMHC structures, and all the extracted entries for the second one. This corresponds to two SSL training scenarios, one where the network sees only pMHC structures, and another where it also sees TCRα/β-peptide structures (Fig. 3C).

### 3D-CNN

Before applying 3D-CNN, we superimposed all pMHC 3D models on the MHC structures, as MHC structures are highly-conserved. This also avoids the rotation-non-equivariant problem of 3D-CNNs. In order to generate consistent 3D grids for each 3D model, all the selected 3D models were aligned using GradPose[69]. The principal component analysis (PCA) of the peptides coordinates was then calculated and the X, Y and Z axis of each PDB file were translated to match with the three principal components, to make sure the axis of the grid would sit along the axis of biggest variation in the peptides' backbone.

For the grids featurization, the PDB files of the aligned pMHC 3D models were then processed with DeepRank software[70] to generate 3D grids as described in Renaud et al.[23]. Each grid was set to have a shape of

35 x 30 x 30 Å and a resolution of 1 Å. Only residues of each chain within 8.5 Å from the other chain are considered "interface" residues and kept for feature calculation. Features were then generated for each atom position and mapped to the grid points with a Gaussian mapping as described in Renaud et al.[23]. In addition to DeepRank's default features, we added an Energy of Desolvation feature calculated as in Dominguez et al.,[71] as the difference in the solvation energy between the bound and unbound complex. We have also replaced DeepRank's default PSSM feature used for sequence embedding with the lighter skip-gram residue embedding taken from Phloyphisut et al.[72]. All the features used are summarized in Table 1. The network architecture is described in detail in the Supplementary Note 4.

### GNN

The PDB files were converted into PPIs in the form of residue-level graphs using DeepRank2 package[73,74]. The latter is a Python package inherited from our previous package DeepRank[70] and that offers a complete framework to learn PPI patterns in an end-to-end fashion using GNNs. For a given PDB file representing a pMHC complex, we defined the interface graph as all the residues involved in intermolecular contacts, i.e. the residues of a given chain having a heavy atom within an 15 Å distance cutoff of any heavy atom from the other chain. These contact residues formed the nodes of the graph. Edges were defined between two contact residues presenting a minimal atomic distance smaller than 15 Å. Node and edge residue-level features used are summarized in Table 2. The network architecture is described in detail in the Supplementary Note 4.

### EGNN

The PDB files were converted into PPIs in the form of residue-level graphs. For a given PDB file representing a pMHC complex, we defined the interface graph as all the residues which carbon α atoms fall within a 10 Å radius of any peptide's residue carbon α. The only features that were used for residues are the residue type, coordinates, and a binary label representing whether the residue comes from the peptide or the MHC. Edges were defined using a k-nearest neighbors (KNN) procedure, where a node is connected to 10 of its closest neighbors. A binary edge attribute distinguishes between cross-chain edges and inter-chain edges. Node and edge features used are summarized in Table 3. The EGNN network is trained in a supervised manner and a self-supervised manner (i.e., 3D-SSL). The details are described in Supplementary Note 4.

### Sequence-based methods

The MLP and re-trained MHCFlurry 2.0[11] predictors use the BLOSUM62 substitution matrix[75] for amino acid representation. The MHC allele is encoded as a 37-residue pseudosequence[13], and the peptide is represented in three 15-residue formats[11]: left-aligned, centered, and right-aligned. Peptides shorter than 15 residues are padded with placeholder residue X. These encoded sequences are combined into an 82×21 matrix (45 × 21 for the peptide and 37 × 21 for the MHC pseudosequence).

The original script from the MHCFlurry 2.0 work[11] was used to build a combined ensemble made of several neural networks. For each data configuration, the only parameters modified from the MHCflurry 2.0 script were the data input and output paths. The original

A

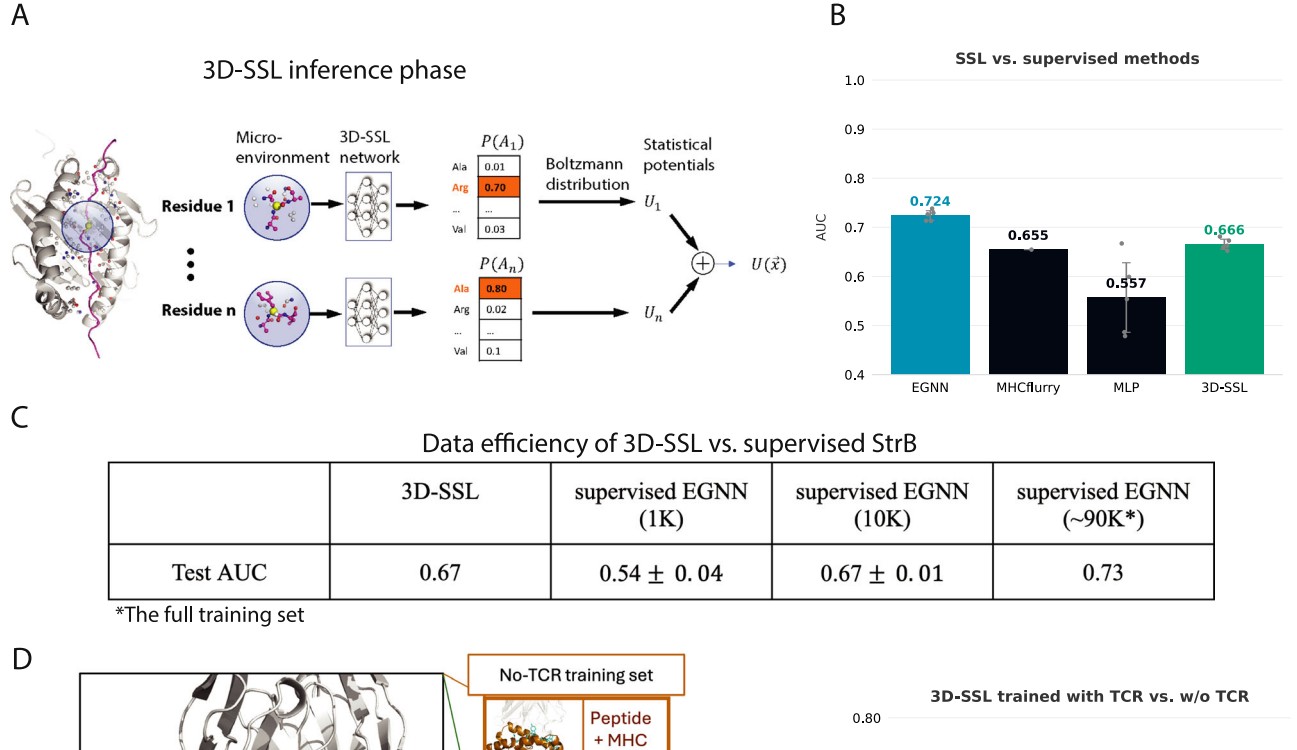

B

C

Data efficiency of 3D-SSL vs. supervised StrB

|  | 3D-SSL | supervised EGNN (1K) | supervised EGNN (10K) | supervised EGNN (~90K*) |
|---|---|---|---|---|
| Test AUC | 0.67 | $0.54 \pm 0.04$ | $0.67 \pm 0.01$ | 0.73 |

*The full training set

D

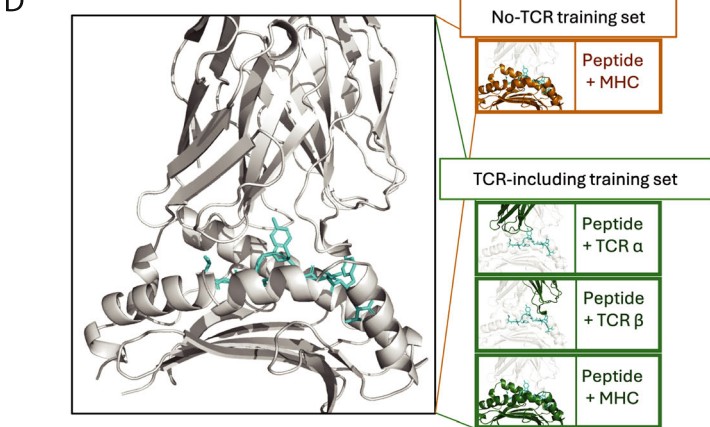

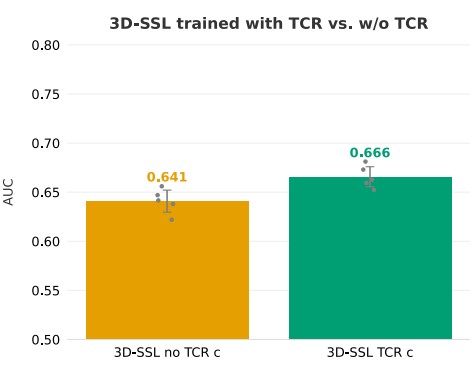

**Fig. 3 | Self-supervised learning approach (3D-SSL) overview and performances. A** the inference stage of 3D-SSL. 3D-SSL is an EGNN network. It takes the micro-environment of a masked peptide residue as input, predicts the probability of amino acid identities of the masked residue, then converts the probability into statistical potential using Boltzmann distribution as the energy contribution of this micro-environment. For the training stage of 3D-SSL, see Supplementary Fig. 2. **B** Comparison of 3D-SSL (i.e., unsupervised EGNN) with the same EGNN network trained in a supervised manner and with SeqB approaches, on the allele-clustered dataset. Replicas are marked as gray dots. There are no allele overlaps between the 3D-SSL training set and the allele-clustered test set. **C** Data efficiency of 3D-SSL against supervised EGNN, our top StrB method. The same EGNN architecture is used by 3D-SSL and supervised EGNN (see Supplementary Note 4). We evaluate the effect of training data size (1 K, 10 K, ~90 K) on supervised EGNN performance. The allele-clustered BA dataset is randomly sampled for training the supervised EGNN. AUCs are averaged over 5 runs to account for the potentially unrepresentative subsets. Error bars show the standard deviation between the five replicas. **D** 3D-SSL performance in terms of AUC when trained with and without peptide-TCR structures, as schematized on the left. We repeated the experiments five times to be sure the difference between training with TCRs and without TCRs is not caused by randomness (error bars for standard deviations shown in black, replicas shown as gray dots).

---

script is available on https://github.com/openvax/mhcflurry/blob/master/downloads-generation/models_class1_pan/GENERATE.sh. The MLP we used consists of a simple architecture built using PyTorch 2.13.

For more detailed information about the inputs, the MLP architecture and the MHCFlurry retraining, refer to Supplementary Note 4.

### *HBV* case study

Immunopeptidomics data was obtained by liquid chromatography tandem mass spectrometry (LC-MS/MS) as described in Kessler et al.[44]. In short, monocyte-derived dendritic cells (moDCs) from 6 healthy donors were pulsed with 12 prototype hepatitis B virus (*HBV*) synthetic long peptides (SLPs) in combination with a maturation stimulus for 22 hours. *HLA-I*-peptide complexes were immunoprecipitated and subjected to data-

dependent acquisition LC-MS/MS analysis. Data was analyzed in PEAKS with FDR5. Obtained spectra from cross-presented, SLP-derived peptides were manually validated by two MS experts. For the values reported in Fig. 4B, NetMHCpan 4.1b from the web server[10] and MHCflurry2.0 locally installed[11] were used to predict the peptides' binding to *HLA-A\*02:01* with default options.

### Statistics and reproducibility

Statistical analyses were conducted using Python libraries including scipy and scikit-learn. The primary metric for evaluating model performance was the AUC, and the AUPRC was used as a secondary metric.

For all neural network models except MHCflurry, experiments were repeated 5 times with different random initializations to account for

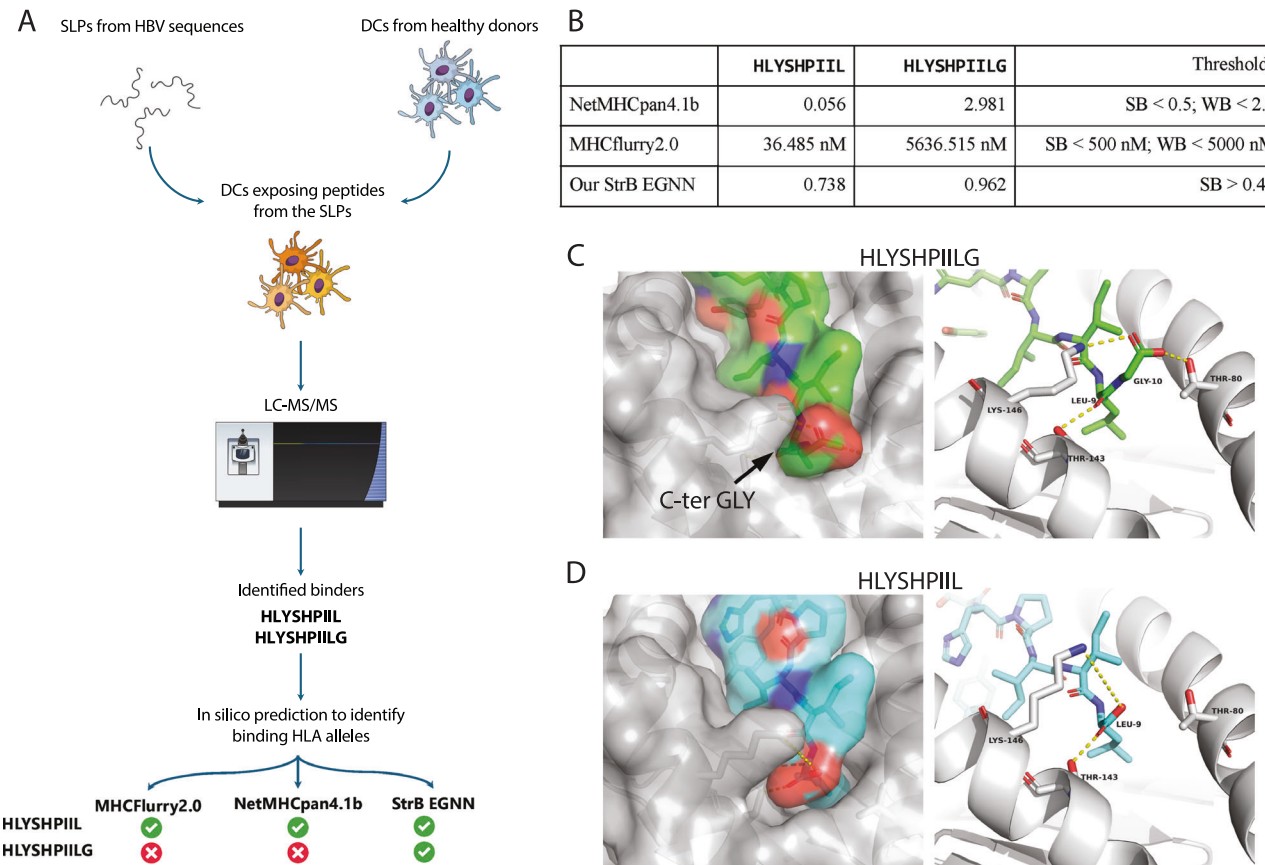

| | HLYSHPIIL | HLYSHPIILG | Thresholds |
|---|---|---|---|
| NetMHCpan4.1b | 0.056 | 2.981 | SB < 0.5; WB < 2.0 |
| MHCflurry2.0 | 36.485 nM | 5636.515 nM | SB < 500 nM; WB < 5000 nM |
| Our StrB EGNN | 0.738 | 0.962 | SB > 0.41 |

**Fig. 4 | Case study on two peptides in *HBV* vaccine research. A** *HBV* case study experiment. Sequences isolated from *HBV* have been used to synthesize SLPs. The SLPs were then added to DCs cultures from 6 healthy donors. Samples were then analyzed by liquid chromatography tandem mass spectrometry (LC-MS/MS), leading to the identification of peptides presented on the surface of DCs. The peptides were then fed into NetMHCpan4.1b, MHCflurry2.0 BA and our StrB GNN and EGNN. Only our StrB methods successfully identified both peptides as binders. Figure created in the Mind the Graph platform www.mindthegraph.com. **B** Outputs and cutoffs from predictors tested. The GNN and EGNN here are trained on the whole ~100 K binding affinity data. "SB" stands for Strong Binder, for which the generally used IC50 cutoff is 500 nM and "WB" stands for Weak Binder, for which the generally used IC50 cutoff is 5000 nM. **C, D** Structural models for HLYSHPIILG (green) and HLYSHPIIL (cyan). Expected polar interactions and hydrogen bonds are shown as yellow dashed lines. Structural models are generated by PANDORA[37]. Visualizations are done in PYMOL[77].

variability. Results are reported as mean AUC values with error bars representing the standard deviation across these 5 replicates. MHCflurry, being an ensemble method, handles separate validation sets internally and collects network outputs, so no standard deviation could be retrieved for its results.

The binding affinity dataset used for training and evaluation consisted of 100,178 data points covering 110 HLA alleles. This dataset was split into training (90%) and test (10%) sets for the shuffled configuration, maintaining similar distributions of positive and negative cases in each set. For the allele-clustered configuration, approximately 10% of data from each HLA gene was selected for the test set based on MHC pseudosequence clustering.

For the 3D-SSL, a separate dataset of 1245 X-ray structures from TCRpMHC complexes was used for training. The performance comparison between 3D-SSL trained with and without TCR structures was repeated 5 times to ensure the observed difference was not due to random variation.

In the data efficiency experiment comparing supervised EGNN with 3D-SSL, subsets of 1 K, 10 K, and ~90 K data points were used for training the supervised model. AUCs were averaged over 5 runs for each subset size to account for potential sampling biases.

The *HBV* case study used immunopeptidomics data obtained from 6 healthy donors, with peptides manually validated by two mass spectrometry experts. While this case study provides valuable insights, it should be noted that the sample size is limited and further validation with larger cohorts would be beneficial for broader generalization.

## Data availability
BA data, 3D models, allele clusters, trained networks and network outputs are available on Zenodo[76] (https://doi.org/10.5281/zenodo.10666413). All the graphs shown in this publication have been plotted directly using such data.

## Code availability
The code for data generation, featurization and training is available at https://github.com/DeepRank/3D-Vac and on Zenodo at https://zenodo.org/records/13820276 (https://doi.org/10.5281/zenodo.13820275).

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

## Acknowledgements
L.X. and D.F.M. are supported by the Hypatia Fellowship from Radboudumc (Rv819.52706), Hanarth Fond and the Kika grant (grant number 454). G.C., S.G., D.L.B., P.L. and N.R. are supported by the Netherlands eScience Center under grant number NLESC.OEC.2021.008. This work was also supported by the NVIDIA Academic Hardware Grant Program. We thank SurfSara for their generous GPU and CPU supercomputing grants (EINF-2380 and EINF-4578). 3D-SSL development was supported by NWO-XS (OCENW.XS23.2.130). We sincerely thank the funding through the AI-For-Health program at Radboudumc for H.S. and D.L. We thank the Erasmus fellowship to D.L. We also thank Farzaneh Meimandi Parizi and Coos Baakman, the main developers of the preprocessing part of the DeepRank2 package, and Sven van der Burg, who helped with the refactoring and the maintenance of the package in the first phase of the project. We thank Prof. Peter-Bram 't Hoen for the constructive scientific discussions and advice.

## Author contributions
Conceived the study: L.X., D.F.M. Prepared the data: D.F.M., D.L., H.S. 3D-CNN experiments: D.F.M., D.T.R., H.S. GNN experiments: G.C., C.L. EGNN and 3D-SSL experiments: T.R. MHCFlurry and MLP: D.L., D.F.M. *HBV* data collection: A.L.K., S.I.B. Project supervision and coordination: L.X., E.B. Manuscript writing and editing: D.F.M., G.C., L.X., S.G., P.L., D.L.B., D.L., A.L.K., S.I.B., E.B. Figures: D.F.M., G.C., L.X., N.R. All authors read and contributed to the manuscript. Giulia and Dario contributed equally.

## Competing interests
The authors declare no competing interests.
