## [Transparent Peer Review file · Communications Biology]

Geometric deep learning improves generalizability of MHC-bound peptide predictions

Corresponding Author: Dr Li Xue

Version 0:

Reviewer comments:

Reviewer #1

(Remarks to the Author)

The authors present a novel method to predict MHC-peptide binding through geometric deep learning. In this work, the hypothetical/predicted MHC-peptide complex structures were first constructed, and then interface graphs were derived. Finally, the binding prediction task is converted to a graph classification problem. Overall, the manuscript is well-written and the experiments are thorough. However, several problems should be addressed before publication.

Major:

1. The authors claimed that self-supervised learning (SSL) is data-efficient and powerful since it achieves an AUC of 0.67 using 1245 structures while supervised EGNN obtains an AUC of 0.54 and 0.67 using 1K and 10K data points, respectively. This seems a bit far-fetched. In general, using SSL for zero-shot prediction is challenging since the model has not been exposed to the labeled data (binding affinity in this case). Therefore, SSL typically requires more data. In my opinion, SSL outperformed supervised learning using the same amount of data may be attributed to the fact that SSL used native complex structures while supervised EGNN used predicted complex structures from PANDORA. I am inquisitive about the performance of SSL using predicted complex structures.
2. The authors claimed that the structure-based method they proposed can improve generalizability, and they adopted data separation using allele clustering, which is good. However, the case study chose the HLA-A*02:01, which is a seen HLA in the training set. This makes the case study slightly unconvincing. Besides, are the peptides HLYSHPIIL and HLYSHPIILG also seen (i.e., similar peptides exist) in the training set of the HLA-A*02:01 allele?
3. Considering that EGNN outperforms GNN in the main results (Figure 2), why did the authors report the results of GNN instead of EGNN in the case study?

Minor:

1. What do SB and WB mean in Figure 4B?
2. What does "Used for pooling" mean in Table 3? Please provide more details of the pooling strategy.
3. The paragraph for explaining GNN describes that "Edges were defined between two contact residues from distinct chains". However, the explanations of the edge features describe that "Boolean indicating whether the edge connects nodes belonging to the same chain (1) or separate chains (0)". This is quite contradictory and confusing.
4. Can the authors provide results that retain three significant figures in the histogram in Figures 2B and 2C? Since I found that the results of CNN and GNN are the same in Figure 2B (0.86 for both methods) while the heights of the bars in the histogram are slightly different.
5. Typo: "AUCPR" in the "StrB predictors demonstrate greater generalizability than SeqB on distant alleles" section; "trained on training on" in the "Self-supervised learning demonstrates superior data efficiency" section.
6. Please add the citation for ProtCID in the "Training data for 3D-SSL" section. Please also cite a related work (<https://doi.org/10.7554/eLife.93695>) in the introduction (e.g., in the sentence "The booming advances of geometric deep learning ...") which also explores geometric deep learning for protein representation.

Reviewer #2

(Remarks to the Author)

Dear Editorial Team and authors,

I had the pleasure of reviewing the manuscript titled “Improving Generalizability for MHC-I Binding Peptide Predictions Through Geometric Deep Learning” for consideration in Communications Biology. The manuscript is well suited to the journal’s audience and standards. It steers the work on MHC-I binding prediction in a vital direction by emphasizing the importance of generalizability testing and data efficiency, which mirrors real-world applications of MHC-I binding models. The generalizability testing highlighted the limitations of sequence-based prediction methods, which are as biased as their input data, including the representation of amino acids, alleles, and sequence redundancy. To address these issues, the manuscript introduces a structure-based prediction method that learns the rules of interfacing with MHC-I binding complexes more efficiently and generalizably, showing considerable success with distant, unseen alleles. Additionally, the manuscript’s exploration of self-supervised learning under low-sample conditions further enhances its contributions to meet real-world utilization scenarios.

The work in the manuscript provides a much-needed correction in the field of MHC-I binding prediction, where models often risk memorizing input data and yet perform well due to testing scenarios that do not account for generalizability. In addition to enhancing adaptability to real-world testing scenarios, the manuscript addresses the issue of low training data availability for less prevalent alleles. Although my expertise primarily lies in sequence-based protein-protein interaction (PPI) methods and their machine-learning biases, my scientific assessment of the details concerning the structural-based method and the ML handling of the structural information finds them to be solid and well-conceived. The case study, while very specific and small in size, provides thoughtful reasoning about the critical methodological differences that enable the proposed method to perform better in that particular scenario.

I have a few comments that should further solidify this elegant piece of work. The comments should not take significant time to address though.

[1] [Minor] The statement, “We set out to test the generalization power of StrB over SeqB approaches to unseen data through two different data splitting configurations of the same binding affinity dataset: a randomly shuffled one, resembling SOTA-used datasets, and an allele-clustered one,” should be revisited. Randomly split data are not a means of assessing generalizability, as the authors themselves discussed and specifically used allele-clustered testing to overcome this generalizability limitation.

[2] [Important] When publishing ML-applied research, it is crucial to describe the data itself, such as the number of positive and negative instances, feature size, feature encoding/representation, and how the MLP handles paired input. This information is currently buried in the Methods/Supplementary section but should be prominently included in the main text. For example, understanding the features used and the data imbalance provides essential context for assessing the results.

[3] [Minor] A non-specialized reader would benefit from the mention of the abbreviation EGNN in the main text rather than just in the Methods/ Supplementary material.

[4] [Important] How does the designed MLP method ensure performance comparable to state-of-the-art SeqB methods? An arbitrary model design might underperform, rendering any comparison moot. A detailed discussion justifying the choice of the MLP method, similar to the discussion about the MHCflurry model differences (the one used compared to the benchmarking performance published for HMCFlurry), would lend credibility to the results. Along the same line, the MHCflurry performance variation discussion needs to mention how different the performance of the two models is; currently, it is only stated that they are different.

[5] [Important] Considering the manuscript focuses on real-world testing scenarios, I encourage the authors to revisit their performance metrics utilized in the manuscript. While I appreciate the use of the AUCPR curve in the Supplementary, this alone is insufficient here because the positive-to-negative data ratio is nearly 1:1, making the AUCPR very close to the general AUC. Assessing the initial (non-filtered) dataset’s positive and negative ratios and then adjusting the testing set during AUCPR reporting could provide a more realistic metric.

[6] [Important] In Figures 2.B, C, and D, the performance differences between the proposed method and other methods appear small. Statistical significance testing might be required here to convince the reader of the results’ credibility.

[7] [Minor] The statement, “Innovatively, we convert the probabilities into a statistical potential using a Boltzmann distribution,” may confuse non-specialized readers as to why this step is innovative without further explanation. Clarifying the importance of this step and whether it handles conditional probabilities in an autoregressive manner would make the general reader understand and appreciate the work.

[8] [Minor] The reader would benefit from additional elaboration or a citation to understand why “predicting the Binding Affinity (BA) of MHC-II is challenging,” as stated at the end of the Discussion.

[9] [Minor] In Supplementary, “peptide 45-mer representation” gives the illusion that the input is of length 45AA, not the actual 15 amino acid, since the -mer is typically reserved in the context of protein to indicate the length.

[10] [Minor] The manuscript's language is generally smooth; however, some phrases like “avoid resorting to data augmentation”, “until reaching the closest possible value to 10% of the data”, and “ubiquitous paradigm” might cause confusion. These could be clarified, possibly with the assistance of an AI tool like ChatGPT, to ensure the writing is unambiguous. This is a non-essential suggestion but can enhance clarity, especially for readers who are non-native English speakers.

Thank you for compiling this work, and I wish you a swift path to publication.

Best regards,
Fatma-Elzahraa Eid
Senior Machine Learning Scientist
Broad Institute of MIT and Harvard

Reviewer #3

(Remarks to the Author)

This is a robust study comparing new methods for structure-based deep learning on peptide-MHC interactions to previous sequence based methods. The authors have attempted to make this a fair comparison by analyzing the same data although it appears that of the sequence based methods only MHCFlurry2.0 is available to the authors.

The authors generated pMHC models with PANDORA and developed CNN/GNN/EGNN based methods to analyse the 3D data showing better performance in predicting binding compared to sequence based methods. They then use the EGNN to test self-supervised approaches using much smaller training sets.

My main concerns/queries focus on the utility of the 3D-SSL approach and whether it will be generalizable to other interactions.

1) Does 3D-SSL require high quality crystal structure data? How does performance compare if it is trained on 1k/10k/90k PANDORA models?

2) For the specific question of interest here - predicting pMHC interactions - does combining all training datasets with the 3D-SSL (pMHC,pTCR,90k PANDORA) have superior performance to the EGNN alone?

Minor point

-define GNN/EGNN the first time they are used.

Version 1:

Reviewer comments:

Reviewer #1

(Remarks to the Author)

All of my concerns have been addressed.

Reviewer #2

(Remarks to the Author)

Dear Editorial Team and Authors,

I had the pleasure of reading the authors' rebuttal to the initial reviewing round and found their responses to be thoughtful and educational. The authors did an excellent job addressing my comments and demonstrated a comprehensive approach to even minor suggestions. I have no further concerns regarding the work, except for Point #5. The authors clarified the source of my concern, which was related to the nature of the data.

To ensure clarity for non-specialist ML readers, I believe it is important to highlight in the main text that models trained on these datasets that are near-balanced may not generalize well to scenarios where negative data points significantly outnumber positive ones. This difference might affect the AUPRC performance.

Editors, to avoid delaying the publication of this work further, I waive the need to review the manuscript after the authors' reconsideration. If the authors can include a clarifying statement in the main text or include an experiment where they report AUPRC when the number of negative data points in the test set is varied, please proceed with the publication.

Thank you for providing such a comprehensive and objective rebuttal. I wish your work a swift publication and hope it significantly impacts research in this area.

Fatma Elzahraa Eid
Senior Machine Learning Scientist
Broad Institute of MIT and Harvard

Reviewer #3

(Remarks to the Author)

Thank you for clarifying my queries. I am convinced this is a useful advance for this field but please ensure the final manuscript does not overstate the potential impact for non-MHC PPIs that are not as well studied (e.g. <1000 crystal structures).

Rebuttal Letter

We thank all the reviewers for the very insightful and constructive feedback, which are undoubtedly helping us improve the quality of our work. We reproduced the reviewers' comments below in black, and our point-to-point responses in green. We also highlighted all changes in yellow in the manuscript.

In addition to addressing the reviewers' comments, we have made several changes not linked to any specific comment to correct unnoticed errors, clarify details, and enhance readability.

1- We have noticed a few cases were not being properly processed by some of the used algorithms, causing the final size of data available to each algorithm slightly different. We have reported this discrepancy in the "Binding affinity data collection" of the methods as such:

"Due to issues in the data processing and data featurization, some networks could use a few datapoints less than the total dataset. The final dataset size per model was: 100,178 for CNN and EGNN; 100,069 for GNN; 100,090 for MHCflurry and MLP. As the difference in datasets amounts for about 0.1% of the data and it is proportionally distributed in train and test sets, we expect it to not influence the results."

We have also added the following sentence to the Methods "Data clustering" subsection:

"Note that the dataset includes 37 HLA-E cases from two alleles, making clustering unfeasible, thus they were included in the training/validation set."

We will release the csv files with the exact cases used for each network on zenodo, together with the rest of the data mentioned in the "Data & Code Availability Statement".

2- We have corrected Figure 2A's flowchart GNN/EGNN image, as it used to show a two-graph approach that was used in early versions of DeepRank2 but ultimately was not used for this work

3- We realized the labels in Figure 2D were in the wrong order, leading the AUCs for incorrect alleles to be reported for some alleles labels. We have now updated the figure with the correct labels and the results from replicate experiments (see our response to Reviewer #2, comment #6).

4- As the work that led to the identification of the HBV case study peptides is now online on researchsquare, we have updated its reference (Kessler *et al.*, reference 43).

Reviewer #1:

The authors present a novel method to predict MHC-peptide binding through geometric deep learning. In this work, the hypothetical/predicted MHC-peptide complex structures were first constructed, and then interface graphs were derived. Finally, the binding prediction task is converted to a graph classification problem. Overall, the manuscript is well-written and the experiments are thorough. However, several problems should be addressed before publication.

Major:

1. The authors claimed that self-supervised learning (SSL) is data-efficient and powerful since it achieves an AUC of 0.67 using 1245 structures while supervised EGNN obtains an AUC of 0.54 and 0.67 using 1K and 10K data points, respectively. This seems a bit far-fetched. In general, using SSL for zero-shot prediction is challenging since the model has not been exposed to the labeled data (binding affinity in this case). Therefore, SSL typically requires more data. In my opinion, SSL outperformed supervised learning using the same amount of data may be attributed to the fact that SSL used native complex structures while supervised EGNN used predicted complex structures from PANDORA. I am inquisitive about the performance of SSL using predicted complex structures.

We thank the reviewer for pointing out the unclear message. In fact, 3D-SSL indeed is trained on more datapoints (about 239,000 3D microenvironments in 1245 X-ray structures) than the EGNN (~90K binding affinity data). We have expanded the following explanation to the *Training* paragraph about 3D-SSL:

“At training time, a random 20% of the residues in the input structure is masked each epoch, i.e. set to a dedicated 'unknown' token, and every masked residue constitutes a separate input datapoint. In this way, every residue in each pMHC structure (around 192 residues) is essentially treated as a separate datapoint, effectively increasing the training set. The EGNN network is then trained to predict the probability of the 20 types of amino acids for each masked residue based on its spatial neighbors.”

We have also edited the following sentence to remove the focus from the supervised vs. unsupervised learning and shift it to the data as well:

“To further estimate the data efficiency of 3D-SSL against StrB methods, we trained the supervised EGNN on small subsets of the BA training set to estimate the amount of 3D models needed for the supervised network to outperform the self-supervised learning on X-ray structures”

We have also expanded the Supplementary Figure 2 caption:

“During the training phase of 3D-SSL, each 3D structure is converted into a residue-level graph, and 20% of its nodes are randomly masked. The graph is then fed

into the EGNN, which returns a residue identity probability for the masked node. Essentially, every masked residue is treated as a separate data point, effectively increasing the training set to the total number of residues in the training structures. The network is trained for enough epochs to make sure that every residue is masked multiple times.”

Finally, to explore the dependence of 3D-SSL performance on the quality of the structures used for training, we added a Supplementary Note (now Supplementary Note 2) where we show 3D-SSL performances when trained on different sets, including only X-rays, increasing amounts of only PANDORA models, and X-rays plus PANDORA models. Our results indeed show that 3D-SSL requires high-quality X-ray structures.

2. The authors claimed that the structure-based method they proposed can improve generalizability, and they adopted data separation using allele clustering, which is good. However, the case study chose the HLA-A*02:01, which is a seen HLA in the training set. This makes the case study slightly unconvincing. Besides, are the peptides HLYSHPIIL and HLYSHPIILG also seen (i.e., similar peptides exist) in the training set of the HLA-A*02:01 allele?

We thank the reviewer for pointing out this unclarity.

In this experiment, differently from the allele-clustered experiment, we retrained our EGNN (see the next comment) on all the available data, without sparing a test set, and we then compared it with the state-of-the-art, pre-trained versions of NetMHCpan4.1b and MHCflurry2.0 as released by the respective authors. For this reason, overlaps with the training set are to be expected, as they are not avoidable in real-case scenario. The HLYSHPIIL peptide is present in all of the training sets, while the HLYSHPIILG peptide is not. We have extended the following sentences to clear this confusion:

“The HL peptide has been reported numerous times in literature as an HLA-A02 binder, thus it is present in the training set of NetMHCpan4.1, MHCflurry 2.0 and our EGNN. The HG peptide instead is not present in any of the training sets to the best of our knowledge, as it was reported for the first time in Kessler *et al.*”

“We first trained our EGNN on all the data available in our dataset, splitting it in train and validation only without any test set, in order to use as much data as possible for training. Then, we generated 3D models of the pMHC complex between the HL/HG peptides and HLA-A*0201, and our trained EGNN networks to make separate predictions. Both of our networks were able to predict both peptides as binders with high confidence scores (**Fig. 4B**).”

3. Considering that EGNN outperforms GNN in the main results (Figure 2), why did the authors report the results of GNN instead of EGNN in the case study?

We have now trained the EGNN on all our available data as we did with the GNN and added its predictions to Fig. 4B. The EGNN shows similar performances to the GNN, predicting both peptides to be binders. As such, the main conclusions are unchanged, and we have replaced the GNN with the EGNN in the test-case study to avoid overcomplicating the story.

Minor:

1. What do SB and WB mean in Figure 4B?

We added the following explanation in the figure caption:

““SB” stands for Strong Binder, for which the generally used IC₅₀ cutoff is 500nM and “WB” stands for Weak Binder, for which the generally used IC₅₀ cutoff is 5000nM.”

2. What does “Used for pooling” mean in Table 3? Please provide more details of the pooling strategy.

We have extended the explanation in the table to the following:

“Boolean indicating whether the node is the peptide (1) or the protein (0). Used during pooling to identify which residues outputs to sum together (i.e. the ones with entity=1, corresponding to the peptide)”

3. The paragraph for explaining GNN describes that “Edges were defined between two contact residues from distinct chains”. However, the explanations of the edge features describe that “Boolean indicating whether the edge connects nodes belonging to the same chain (1) or separate chains (0)”. This is quite contradictory and confusing.

Edges were defined between two contact residues presenting a minimal atomic distance smaller than 15 Å. Thus, an edge can connect both two nodes belonging to the same chain and two nodes belonging to two distinct chains. After having carefully checked the code, the text has been updated accordingly to remove the confusion: “Edges were defined between two contact residues presenting a minimal atomic distance smaller than 15 Å”.

4. Can the authors provide results that retain three significant figures in the histogram in Figures 2B and 2C? Since I found that the results of CNN and GNN are the same in Figure 2B (0.86 for both methods) while the heights of the bars in the histogram are slightly different.

We now use three digits after the periods instead of two for all the figures.

5. Typo: “AUCPR” in the “StrB predictors demonstrate greater generalizability than SeqB on distant alleles” section; “trained on training on” in the “Self-supervised learning demonstrates superior data efficiency” section.

Thanks, we corrected the typo.

6. Please add the citation for ProtCID in the “Training data for 3D-SSL” section. Please also cite a related work (<https://doi.org/10.7554/eLife.93695>) in the introduction (e.g., in the sentence “The booming advances of geometric deep learning ...”) which also explores geometric deep learning for protein representation.

We added the suggested references and mentioned them in the text accordingly.

Reviewer #2 (Remarks to the Author):

Dear Editorial Team and authors,

I had the pleasure of reviewing the manuscript titled “Improving Generalizability for MHC-I Binding Peptide Predictions Through Geometric Deep Learning” for consideration in Communications Biology. The manuscript is well suited to the journal’s audience and standards. It steers the work on MHC-I binding prediction in a vital direction by emphasizing the importance of generalizability testing and data efficiency, which mirrors real-world applications of MHC-I binding models. The generalizability testing highlighted the limitations of sequence-based prediction methods, which are as biased as their input data, including the representation of amino acids, alleles, and sequence redundancy. To address these issues, the manuscript introduces a structure-based prediction method that learns the rules of interfacing with MHC-I binding complexes more efficiently and generalizably, showing considerable success with distant, unseen alleles. Additionally, the manuscript’s exploration of self-supervised learning under low-sample conditions further enhances its contributions to meet real-world utilization scenarios.

The work in the manuscript provides a much-needed correction in the field of MHC-I binding prediction, where models often risk memorizing input data and yet perform well due to testing scenarios that do not account for generalizability. In addition to enhancing adaptability to real-world testing scenarios, the manuscript addresses the issue of low training data availability for less prevalent alleles. Although my expertise primarily lies in sequence-based protein-protein interaction (PPI) methods and their machine-learning biases, my scientific assessment of the details concerning the structural-based method and the ML handling of the structural information finds them to be solid and well-conceived. The case study, while very specific and small in size, provides

thoughtful reasoning about the critical methodological differences that enable the proposed method to perform better in that particular scenario.

I have a few comments that should further solidify this elegant piece of work. The comments should not take significant time to address though.

[1] [Minor] The statement, “We set out to test the generalization power of StrB over SeqB approaches to unseen data through two different data splitting configurations of the same binding affinity dataset: a randomly shuffled one, resembling SOTA-used datasets, and an allele-clustered one,” should be revisited. Randomly split data are not a means of assessing generalizability, as the authors themselves discussed and specifically used allele-clustered testing to overcome this generalizability limitation.

We rephrased the sentence to clarify that the shuffled configuration was used as a baseline for testing the models, while the configuration with allele-clustered data was used to assess their generalization power. Specifically, we wrote: “We tested the StrB over SeqB approaches on unseen alleles using two data configurations: a randomly shuffled data configuration as the control baseline, resembling the SOTA-used data separation schemes, and an allele-clustered configuration to evaluate their generalization power.”

[2] [Important] When publishing ML-applied research, it is crucial to describe the data itself, such as the number of positive and negative instances, feature size, feature encoding/representation, and how the MLP handles paired input. This information is currently buried in the Methods/Supplementary section but should be prominently included in the main text. For example, understanding the features used and the data imbalance provides essential context for assessing the results.

We made these points more clear by adding the following sentences in the Results section:

- “To evaluate this hypothesis, we use ~100K peptide-MHC binding affinity data covering 110 HLA alleles and peptides lengths spanning between 8 and 21 residues (**Fig. 1A**, see details in Methods).”
- “For all configurations, pMHC 3D models were processed and featurized using various software tools to generate encoded sequences, 3D grids, or graphs, depending on the approach. For a comprehensive overview of the featurized data characteristics, including feature size, encoding, and representation, refer to the Methods section (**Tables 1-3**).”
- “Binders and non-binders were represented at 44% and 56% in each set, respectively.”

Added in Methods section:

- “The MLP and re-trained MHCFlurry 2.0¹¹ predictors use the BLOSUM62 substitution matrix⁷⁶ for amino acid representation. The MHC allele is encoded as a 37-residue pseudosequence¹³, and the peptide is represented in three 15-residue formats¹¹: left-aligned, centered, and right-aligned. Peptides shorter than 15 residues are padded with placeholder residue X. These encoded sequences are combined into an 82x21 matrix (45x21 for the peptide and 37x21 for the MHC pseudosequence).”
- “For more detailed information about the inputs, the MLP architecture and the MHCFlurry retraining, refer to **Supplementary Note 4.**”

[3] [Minor] A non-specialized reader would benefit from the mention of the abbreviation EGNN in the main text rather than just in the Methods/ Supplementary material.

We added the definition the first time the EGNN term was used, and we did the same for GNN, whose full name was not mentioned in the main text.

[4] [Important] How does the designed MLP method ensure performance comparable to state-of-the-art SeqB methods? An arbitrary model design might underperform, rendering any comparison moot. A detailed discussion justifying the choice of the MLP method, similar to the discussion about the MHCflurry model differences (the one used compared to the benchmarking performance published for MHCFlurry), would lend credibility to the results. Along the same line, the MHCFlurry performance variation discussion needs to mention how different the performance of the two models is; currently, it is only stated that they are different.

We have added the following explanation to the results section “**StrB predictors demonstrate greater generalizability than SeqB on distant alleles**”:

“Considering that MHCflurry2.0 is an ensemble predictor made of multiple MLPs, we designed our MLP to use one representative architecture of individual MLPs in MHCflurry2.0, in order to assess the single MLP contribution compared to the ensemble effect on the predictions performances. Moreover, our MLP serves the purpose of showing how simple it can be for a SeqB method to achieve high performances in a shuffled set configuration, and that this does not guarantee similar performances on unseen alleles.”

[5] [Important] Considering the manuscript focuses on real-world testing scenarios, I encourage the authors to revisit their performance metrics utilized in the manuscript. While I appreciate the use of the AUCPR curve in the Supplementary, this alone is insufficient here because the positive-to-negative data ratio is nearly 1:1, making the AUCPR very close to the general AUC. Assessing the initial (non-filtered) dataset’s

positive and negative ratios and then adjusting the testing set during AUCPR reporting could provide a more realistic metric.

We are afraid there might be a misunderstanding leading the reviewer to think we artificially filtered the data to obtain a balanced dataset (44% positives, 56% negatives), but we did no such filtering. We used all binding affinity data available and only removed very noisy data points. The data happens to be balanced.

On the other hand, we agree with the reviewer that the positive/negative ratio in real life is a crucial matter. The ratios in the data used in this study do not reflect the real biological ratios, but this would be too complex to add in this proof-of-principle work.

[6] [Important] In Figures 2.B, C, and D, the performance differences between the proposed method and other methods appear small. Statistical significance testing might be required here to convince the reader of the results' credibility.

We have performed 5 replicas of each experiment and added the error bars in the figures. The replicas were performed by subsampling 5 different test and validation sets and keeping the test set constant for the shuffled experiment (figure 2B). For the Allele Clustered experiment, we kept the test fixed as it was originally designed (Figure 1) and changed the training and validation set. Due to the structure of MHCflurry, where every model of the ensemble uses a different validation set and the final prediction of the final best models is collected together, we cannot externally define different validation sets without changing MHCflurry's code.

We have added the following sentences to Figure 2 B-C caption:

“For all the networks except MHCflurry, 5 replicas were performed by randomly re-sampling each validation set, and the error bars show the standard deviation between the five replicas. MHCflurry handles the separate validation sets internally and collects the networks' outputs, as such no standard deviation could be retrieved.”

We also added the following sentence to Figures 2D and 3B:

“Error bars show the standard deviation between the five replicas.”

[7] [Minor] The statement, “Innovatively, we convert the probabilities into a statistical potential using a Boltzmann distribution,” may confuse non-specialized readers as to why this step is innovative without further explanation. Clarifying the importance of this step and whether it handles conditional probabilities in an autoregressive manner would make the general reader understand and appreciate the work.

To provide a clearer explanation, we have expanded the explanation on the inference step of 3D-SSL:

“Once the network has been trained on X-ray structures, we use it to predict the peptide binding of our test allele-clustered dataset. The trained network takes a pMHC 3D model as input, masks each residue in the peptide and predicts the probability of the 20 amino acid types at the masked position. The key innovation consists then in converting the predicted probabilities into the binding energy a statistical potential using a Boltzmann distribution. In computational structural biology, Boltzmann distribution is widely used to approximate energies (known as “statistical potentials”) for predicting protein folding and computational docking. In Boltzmann distribution, the frequency of an observed state i is proportional to its energy e_i :

$$p_i \propto e^{-\frac{e_i}{kT}}$$

where k is the Boltzmann constant and T is the temperature.

In traditional statistical potentials, the state i often is a residue-residue contact pair (two-body). Intuitively, if a residue-residue pair is often observed in the PDB structures (i.e., has a high frequency), such interactions are believed to have low energies (the hallmark of native conformations).

In this work, we convert the predicted probabilities from 3D-SSL into the energy contribution using Boltzmann distribution: $e_i = -\log(p_i)$ (for simplicity, we ignored the reference state and set $kT=1$ here). In essence, the 3D-SSL is predicting multi-body statistical potentials. The energy contributions of each peptide residue are then summed up as the binding energy of the whole pMHC structure (Fig. 3A, Methods). We used this final score as a prediction for the binding affinity. Each residue's contribution is treated as an independent event, meaning the conditional probabilities of each residue are considered in a non-autoregressive manner.”

[8] [Minor] The reader would benefit from additional elaboration or a citation to understand why “predicting the Binding Affinity (BA) of MHC-II is challenging,” as stated at the end of the Discussion.

We have added the following sentence to the last paragraph of the discussion:

“In fact, MHC-II have an open-ended binding cleft that allows for longer peptides to bind, thus including peptide flanking regions (PFRs) which are highly variable in length and can affect the pMHC binding affinity⁶³ ”

[9] [Minor] In Supplementary, “peptide 45-mer representation” gives the illusion that the input is of length 45AA, not the actual 15 amino acid, since the -mer is typically reserved in the context of protein to indicate the length.

We have changed “45-mer” to “45-position” to clarify this point, and added the mention that this encoding was originally called 45-mer by the MHCflurry2.0 authors in order not to rename their encoding.

[10] [Minor] The manuscript's language is generally smooth; however, some phrases like “avoid resorting to data augmentation”, “until reaching the closest possible value to 10% of the data”, and “ubiquitous paradigm” might cause confusion. These could be clarified, possibly with the assistance of an AI tool like ChatGPT, to ensure the writing is unambiguous. This is a non-essential suggestion but can enhance clarity, especially for readers who are non-native English speakers.

We thank the reviewer for the kind remark and the care for the accessibility of our work.

- We have changed the sentence including “avoid resorting to data augmentation” to:
“Before applying 3D-CNN, we superimposed all pMHC 3D models on the MHC structures, as MHC structures are highly-conserved. This also avoids the rotation-inequivalent problem of 3D-CNNs.”
- We have changed the sentence with “ubiquitous paradigm” to:
“The message passing framework is commonly used for training GNNs for molecular data.”
- We understand the phrase “until reaching the closest possible value to 10% of the data” might seem ambiguous. As this data separation step is not easy to describe precisely, we provide the precise HLA-allele IDs for the test and training dataset in our data depository to Zenodo.
- We have gone over the whole manuscript trying to smoothen the language as much as possible.

Thank you for compiling this work, and I wish you a swift path to publication.

Best regards,
Senior Machine Learning Scientist

Reviewer #3 (Remarks to the Author):

This is a robust study comparing new methods for structure-based deep learning on peptide-MHC interactions to previous sequence based methods. The authors have attempted to make this a fair comparison by analyzing the same data although it

appears that of the sequence based methods only MHCFlurry2.0 is available to the authors.

The authors generated pMHC models with PANDORA and developed CNN/GNN/EGNN based methods to analyse the 3D data showing better performance in predicting binding compared to sequence based methods. They then use the EGNN to test self-supervised approaches using much smaller training sets.

My main concerns/queries focus on the utility of the 3D-SSL approach and whether it will be generalizable to other interactions.

1) Does 3D-SSL require high quality crystal structure data? How does performance compare if it is trained on 1k/10k/90k PANDORA models?

Yes, 3D-SSL does require high-quality crystal structure data. We have added a new Supplementary Note, renumbered as Supplementary Note 2, where we show the AUC of 3D-SSL trained on different amounts of PANDORA models and X-ray structures. As results clear, PANDORA models provide lower-quality information to the 3D-SSL than the X-ray structures, as even using 40K models allows us to reach an AUC of only 0.64, against the 0.67 achieved by training on X-rays only. Training with X-rays plus PANDORA models does provide for a very small and barely significant performance increase.

2) For the specific question of interest here - predicting pMHC interactions - does combining all training datasets with the 3D-SSL (pMHC,pTCR,90k PANDORA) have superior performance to the EGNN alone?

We have added to the new Supplementary Note 2 an experiment where we trained the 3D-SSL on all the data we could use for it from our datasets (X-ray pMHC and pTCR, and 40K PANDORA models with positive labels (i.e., binding peptides), and achieved a very minor performance increase over the X-rays only (1% improvement). Overall, we envision it will be possible to boost these performances by using the two training strategies in parallel: the EGNN to train on 3D models of labeled binding data, and the 3D-SSL to train on X-ray structures, combining the final predictions together.

Minor point

-define GNN/EGNN the first time they are used.

Thanks, we added their definition the first time they are used.

Rebuttal Letter

We thank the reviewers for their final remarks and their care in reviewing our manuscript. We believe the feedback we received was very constructive and helped us considerably improve the quality of our manuscript. Here we report in green the edits we made to address the reviewers' final concerns.

REVIEWERS' COMMENTS:

Reviewer #1 (Remarks to the Author):

All of my concerns have been addressed.

Reviewer #2 (Remarks to the Author):

Dear Editorial Team and Authors,

I had the pleasure of reading the authors' rebuttal to the initial reviewing round and found their responses to be thoughtful and educational. The authors did an excellent job addressing my comments and demonstrated a comprehensive approach to even minor suggestions. I have no further concerns regarding the work, except for Point #5. The authors clarified the source of my concern, which was related to the nature of the data.

To ensure clarity for non-specialist ML readers, I believe it is important to highlight in the main text that models trained on these datasets that are near-balanced may not generalize well to scenarios where negative data points significantly outnumber positive ones. This difference might affect the AUPRC performance.

Editors, to avoid delaying the publication of this work further, I waive the need to review the manuscript after the authors' reconsideration. If the authors can include a clarifying statement in the main text or include an experiment where they report AUPRC when the number of negative data points in the test set is varied, please proceed with the publication.

Thank you for providing such a comprehensive and objective rebuttal. I wish your work a swift publication and hope it significantly impacts research in this area.

Fatma Elzahraa Eid
Senior Machine Learning Scientist
Broad Institute of MIT and Harvard

We thank the reviewer for the kind words and for the careful review of our work. We have added the following sentence to the results section:

“Note that the AUPRC values reported here may differ in real-world scenarios with highly imbalanced data, where negative instances significantly outnumber positive ones. Our goal is to demonstrate the relative generalization of geometric deep learning over sequence-based methods, though performance may vary with dataset imbalance.”

Reviewer #3 (Remarks to the Author):

Thank you for clarifying my queries. I am convinced this is a useful advance for this field but please ensure the final manuscript does not overstate the potential impact for non-MHC PPIs that are not as well studied (e.g. <1000 crystal structures).

We have modified the abstract from

“This proof-of-concept study highlights structure-based methods’ potential to enhance generalizability and data efficiency, with important implications for data-intensive fields like T-cell receptor specificity predictions, paving the way for enhanced comprehension and manipulation of immune responses.”

To

“This proof-of-concept study highlights structure-based methods’ potential to enhance generalizability and data efficiency, with possible implications for data-intensive fields like T-cell receptor specificity predictions”

Moreover, we added the following sentence to the last paragraph of discussion:

“However, 3D-SSL is trained and tested on pMHC complexes alone and its impact on other protein-protein interfaces (PPIs) needs further extensive investigations. “